# Beyond the new normal: Assessing the feasibility of vaccine-based suppression of SARS-CoV-2

**Madison Stoddard**[1]*, **Sharanya Sarkar**[2], **Lin Yuan**[1], **Ryan P. Nolan**[3], **Douglas E. White**[4], **Laura F. White**[5], **Natasha S. Hochberg**[6,7,8], **Arijit Chakravarty**[1]

**1** Fractal Therapeutics, Cambridge, MA, United States of America, **2** Department of Microbiology and Immunology, Dartmouth College, Hanover, NH, United States of America, **3** Halozyme Therapeutics, San Diego, CA, United States of America, **4** Independent Researcher, United States of America, **5** Department of Biostatistics, Boston University, Boston, MA, United States of America, **6** Department of Epidemiology, Boston University School of Public Health, Boston, MA, United States of America, **7** Department of Medicine, Boston University School of Medicine, Boston, MA, United States of America, **8** Boston Medical Center, Boston, MA, United States of America

* madison.stoddard@fractaltx.com

**Data Availability Statement:** All relevant data are within the paper and its Supporting Information files.

## Abstract

As the COVID-19 pandemic drags into its second year, there is hope on the horizon, in the form of SARS-CoV-2 vaccines which promise disease suppression and a return to pre-pandemic normalcy. In this study we critically examine the basis for that hope, using an epidemiological modeling framework to establish the link between vaccine characteristics and effectiveness in bringing an end to this unprecedented public health crisis. Our findings suggest that a return to pre-pandemic social and economic conditions without fully suppressing SARS-CoV-2 will lead to extensive viral spread, resulting in a high disease burden even in the presence of vaccines that reduce risk of infection and mortality. Our modeling points to the feasibility of complete SARS-CoV-2 suppression with high population-level compliance and vaccines that are highly effective at reducing SARS-CoV-2 infection. Notably, vaccine-mediated reduction of transmission is critical for viral suppression, and in order for partially-effective vaccines to play a positive role in SARS-CoV-2 suppression, complementary biomedical interventions and public health measures must be deployed simultaneously.

## Introduction

The ongoing COVID-19 pandemic, caused by the novel coronavirus SARS-CoV-2, is the health crisis of our lifetimes, placing a burden on global health and economic well-being that would have been inconceivable to us at the start of 2020. The biomedical community has responded with unprecedented speed, and several vaccines are now being rolled out worldwide. The experiences of 2020 have shown that the high basic reproduction number ($R_0$) and capacity for presymptomatic spread make SARS-CoV-2 difficult to control once it is established within a population [1, 2]. Thus, if we are ever to return to a pre-pandemic normal, a focus on SARS-CoV-2 suppression will be critical, and vaccines are expected to play a crucial role in this endeavor.

**Funding:** Fractal Therapeutics provided support in the form of salaries for authors MS and YL, but did not have any additional role in the study design, data collection and analysis, decision to publish, or preparation of the manuscript. The specific roles of these authors are articulated in the 'author contributions' section.

**Competing interests:** I have read the journal's policy and the authors of this manuscript have the following competing interests: MS, LY, and AC are employees and shareholders of Fractal Therapeutics; RN and DW are also shareholders in Fractal Therapeutics. RN is an employee of Halozyme Therapeutics. This does not alter our adherence to PLOS ONE policies on sharing data and materials.

For SARS-CoV-2, an efficacious vaccine might prevent colonization of the nasal cavity (referred to in this work as infection), symptomatic disease, severe disease, mortality or transmission [3]. In the clinical trials for the front-runner vaccines (AstraZeneca/Oxford, Moderna and Pfizer/BioNTech), the primary endpoint is symptom-gated efficacy, where cases are counted as patients presenting with symptoms of COVID-19 and SARS-CoV-2 infection confirmed by nucleic acid amplification test.

The characteristics of SARS-CoV-2 raise a number of potential risks for vaccine-based suppression strategies: waning natural [4] and vaccine-mediated immunity [5], disease transmission by vaccinated individuals [6], age-dependent disease severity [7] and vaccinal immunity [8, 9] and the emergence of resistance [10, 11].

Waning natural immunity, originally proposed as a potential risk for SARS-CoV-2 based on the behavior of other coronaviruses [12, 13] has now been demonstrated for cellular [14] as well as humoral [4, 15] immunity in response to COVID-19. Numerous cases of SARS-CoV-2 reinfection have been proven using direct molecular techniques [16, 17]. The relative infrequency of such events has been shown to be only a consequence of epidemiological dynamics–in a situation where the uninfected population is larger than the previously-infected population, reinfections will make up a small proportion of new infections, but the rate-limiting factor is the percentage of previously-infected individuals and not the duration of immunological protection [18]. The potential for relatively short-term immunity complicates disease control by limiting the duration of natural herd immunity and providing the virus with a foothold even in populations that have previously experienced a high attack rate [19]. If vaccinal immunity were to also wane over similar timeframes, the logistical burden of a vaccine-based suppression strategy for this disease may become insurmountable.

Transmission by vaccinated but colonized individuals poses another risk for vaccine-based suppression of SARS-CoV-2. While prevention of infection (sterilizing immunity) is the norm for many vaccines, there are a number of vaccines (such as the inactivated polio vaccine [20] and the polyvalent pneumococcal vaccine [21]) that merely decrease the potential for onward transmission from colonized but vaccinated individuals. Preclinical studies for a number of COVID-19 vaccine candidates have demonstrated viral infection of the nasal cavity in vaccinated primates [22, 23]. The AstraZeneca clinical study reports a 49.5% reduction in all positive nasopharyngeal swabs in the standard dose vaccinated group with no relative reduction in the number of asymptomatics [6]. Clinical data for the Janssen single-dose vaccine suggests a 66% reduction in symptomatic COVID-19 and a 74% reduction in asymptomatic seroconversion [24]. On the other hand, a CDC study suggests a 90% reduction in risk of infection among working-age adults recently vaccinated with Pfizer or Moderna vaccines [25], while the UK SIREN study similarly reports an 86% reduction in risk of infection among fully vaccinated participants (who largely received the Pfizer vaccine) [26]. Many studies of this nature [27, 28] are limited by the inclusion of voluntary or symptom-gated SARS-CoV-2 test results, which may exaggerate vaccine efficacy against infection if vaccinees are less likely to present with classic COVID-19 symptoms or less likely to seek testing.

The age structure of morbidity and mortality of COVID-19 is further unhelpful in the context of the age-dependence of benefits derived from vaccination. For every two decades of age, the infection fatality rate (IFR) for COVID-19 rises by about one order of magnitude, ranging from 0.013% at 25 years to about 8% at 85 years [7]. Age-dependent reductions in vaccine efficacy have been demonstrated for a number of other diseases [21, 29–31]. For some SARS-CoV-2 vaccines, antibody titers peak lower [9] and decline more rapidly [5] in the over-65 population. Current clinical trials show a high degree of uncertainty for vaccine efficacy in older volunteers. For example, based on clinical data for Moderna's vaccine, efficacy against symptomatic COVID-19 is 95.6% in the 18–64 year-old age group (95% CI 90.6% - 97.9%). In

the over-65 group, vaccine efficacy is reported to be 86.4% with a much wider confidence interval (95% CI 61.4%– 95.5%) [32]. The 95% confidence interval for vaccine efficacy against symptomatic COVID-19 reported by Pfizer is 66.7% to 99.9% for the over-65 age group [8], and the subgroup analysis for the AstraZeneca/Oxford vaccine was not performed due to methodological issues with the trial design resulting in age being confounded with dose schedule [6], leading some countries to recommend the AstraZeneca/Oxford vaccine only in the under-65 population [33].

Widespread, endemic SARS-CoV-2 infection within the population will also create a fertile breeding ground for immune evasion mutants. Viral genomic data collected over the course of the pandemic shows substantial sequence variation for the viral spike protein [34], as well as other surface proteins [35]. Evolutionary modeling conducted by us suggests that SARS-CoV-2 immune evasion mutants with one or two mildly deleterious mutations likely pre-exist in high numbers due to neutral genetic variation, and resistance to vaccines targeting multiple epitopes on the spike protein may occur rapidly under certain conditions [10]. Consistent with these predictions, recent reports have noted the emergence of virus strains (e.g. the South African 501Y.V2 variant) that are capable of complete immune escape from therapeutically relevant monoclonal antibodies, as well as convalescent plasma [36].

Taken together, the risks described above represent a set of real-world constraints for any SARS-CoV-2 suppression strategy that relies primarily on vaccines to bring us back to the 'old normal'. In this study, we use an epidemiological modeling framework to explore the impact of waning natural immunity, limited knowledge of vaccine efficacy against infection and transmission, vaccine refusal, and uncertainty in vaccine efficacy against mortality in individuals older than 65 on the feasibility of SARS-CoV-2 suppression using vaccines. We also estimate the mortality burden of COVID-19 under endemic scenarios managed by partially-effective vaccination schemes. We further extend this work to examine practically useful strategies for vaccine-based suppression of SARS-CoV-2.

## Methods

### SEIRS model

To evaluate strategies for achieving suppression of SARS-CoV-2 and to predict the impact of existing vaccines, we built an SEIRS (susceptible-exposed-infectious-recovered-susceptible) model to account for disease spread, waning natural immunity, and the deployment of a vaccine in some or all of the population. The model has two parallel sets of SEIR compartments representing the vaccinated and unvaccinated populations. Although we recognize that vaccine coverage may be geographically heterogeneous, for the purposes of this model we assume all compartments are well-mixed, meaning that all vaccinated and unvaccinated individuals are in contact and can infect each other. The vaccine may confer a benefit at any of multiple stages: infection, transmission, and disease progression. Thus, the vaccinated population may have a reduced susceptibility to infection ("reduction in risk of infection"), a reduced transmissibility upon infection ("reduction in risk of transmission"), or a reduced likelihood of downstream negative health outcomes upon infection ("reduction in risk of symptoms", "reduction in risk of mortality"). We note that risk of infection and risk of transmission are the determinants of SARS-CoV-2 infection risk and spread, while downstream risks of symptomatic COVID-19 or mortality impact the outcomes of the infected population. Vaccinal immunity is assumed to be life-long or continually boosted such that the degree of vaccinal immunity does not wane or vary in vaccinated individuals. This represents the best-case scenario for vaccine effectiveness; the feasibility of frequent repeat vaccinations and constraints on the vaccine deployment timeline are outside the scope of this paper. Additionally, vaccinated individuals

who become infected ($I_v$) are assumed to develop natural immunity that provides transient protection from reinfection. The ordinary differential equations (ODE)-based SEIRS model is summarized by the following equations:

$$\frac{dS_v}{dt} = -\beta b S_v(cI_v + I_u) + \delta R_v + f\mu - \lambda S_v$$

$$\frac{dE_v}{dt} = -\alpha E_v + \beta b S_v(cI_v + I_u) - \lambda E_v$$

$$\frac{dI_v}{dt} = -\gamma I_v + \alpha E_v - \lambda I_v$$

$$\frac{dR_v}{dt} = \gamma I_v(1-\sigma) - \delta R_v - \lambda R_v$$

$$\frac{dS_u}{dt} = -\beta S_u(cI_v + I_u) + \delta R_u + (1-f)\mu - \lambda S_u$$

$$\frac{dE_u}{dt} = -\alpha E_u + \beta S_u(cI_v + I_u) - \lambda E_u$$

$$\frac{dI_u}{dt} = -\gamma I_u + \alpha E_u - \lambda I_u$$

$$\frac{dR_u}{dt} = \gamma I_u(1-\sigma) - \delta R_u - \lambda R_u$$

Where $S$ represents the susceptible population, $E$ represents the exposed population, $I$ represents the infectious population, and $R$ represents the recovered population. Subscripts $v$ and $u$ delineate the vaccinated and unvaccinated subpopulations, respectively. Model parameters are summarized in Table 1.

We set the vaccine's efficacy against infection to the reported efficacy for the AstraZeneca vaccine, and we estimate the reduction in risk of transmission upon infection to be 9% based on the change in frequency of asymptomatic cases. (See S1 File for detailed calculation.)

**Table 1. Model parameters for SEIRS model.**

| Parameter | Symbol | Value | Source |
|---|---|---|---|
| Latency period | $1/\alpha$ | 3 days | [37] |
| Reproductive number | $R_0$ | 5.7 individuals | [38] |
| Infectious period | $1/\gamma$ | 10 days | [39] |
| Duration of natural immunity | $1/\delta$ | 18 months | [40] |
| Infection fatality rate (IFR) | $\sigma$ | 0.68% | [41] |
| Population birth rate | $\mu$ | 1% annually | [42] |
| Population death rate | $\lambda$ | 0.9% annually | [43] |
| Fraction compliant | $f$ | Variable | |
| Vaccine reduction in risk of infection | $1-b$ | 70% | [24] |
| Vaccine reduction in risk of transmission | $1-c$ | 9% | Estimate |

The contact rate $\beta$ is derived from the relationship between the intrinsic reproductive number $R_0$, the contact rate, and the infectious period:

$$\beta = \gamma R_0$$

According to the CDC, the $R_0$ for SARS-CoV-2 under pre-pandemic social and economic conditions is estimated to be 5.7 based on data from China in early 2020. Although the $R_0$ is commonly held to be in the range of 2–3 based on early estimates, the CDC's study provides an updated estimate based on improved methods designed to address stochasticity in early outbreak dynamics and drawn from a larger set of case reports to address the low numbers of observations that early studies were reliant on [38]. For the purpose of this study, an $R_0$ of 5.7 is used to represent epidemiological conditions under a theoretical return to pre-pandemic activity. In the supplement, the simulations are re-implemented with an $R_0$ of 3.32, in keeping with the results reported by a different systematic meta-analysis of the range of $R_0$s reported in the early stages of the pandemic [44].

In the long-term, the SEIRS system is bistable with two steady-state equilibria: either "suppression," a condition under which yearly SARS-CoV-2 infection rates among the human population stably approach zero, or "endemic disease," a scenario in which SARS-CoV-2 infection and transmission persist at a constant rate at steady-state determined by the effective population contact rate [45]. In this analysis, we further modified the model to explore the tendency toward suppression or endemic disease under a variety of scenarios.

## Model adaptation: Vaccine-mediated reduction in mortality

To address the possibility of a vaccine that prevents COVID-19 symptoms or mortality but has a limited impact on nasopharyngeal SARS-CoV-2 infection, we adapted the model to focus on a mortality-reduction strategy. In this case, we explore the impact of vaccine's efficacy against mortality or symptomatic COVID-19. For these simulations, we track the expected yearly death toll. In the supplement, we explore the impact of these assumptions by running the simulations with no reduction in infection or transmission.

## Model adaptation: Age-dependent vaccine efficacy

Unfortunately, vaccines commonly are less effective in older individuals [21, 29–31], and older individuals are significantly more susceptible to fatal COVID-19 outcomes [7]. To understand the impact of potentially lower vaccine efficacy in the elderly population, we conducted epidemiological modeling to assess the steady-state mortality burden using the SEIRS model. Specifically, we simulated a scenario where vaccine efficacy against mortality in the under-65 age group was fixed at 95%, and vaccine protection for the over-65 age group was varied across a range. We considered the over-65 age group to represent 16.4% of the US population, consistent with current demographic estimates [46]. We calculated a population age structure-adjusted IFR of 3.34% among over-65 Americans and an IFR of 0.16% among under-65 Americans based on age-dependent US IFRs [7].

## Model adaptation: Stacked interventions

We also modified the model to simulate disease dynamics in a scenario where two complementary biomedical or nonpharmaceutical interventions are implemented. In this model, there are four sets of SEIR compartments, representing the four permutations of participation and nonparticipation with the two interventions. The choice to participate in each intervention is assumed to be independent: thus, the fraction of the population participating in both

interventions is $f_1 * f_2$. The population is assumed to be well-mixed with no strategy-switching during the modeled time period. In these scenarios, we track the total number of US infections on a yearly basis at steady-state.

## Results

### Without effective intervention, endemic SARS-CoV-2 spread will be extensive

If vaccines prevent symptom onset but do not prevent infection, SARS-CoV-2 will become endemic in the population, reaching a steady-state yearly number of cases determined by the duration of natural immunity and the contact rate in the population, which determines $R_0$. As shown in S5 Fig, yearly SARS-CoV-2 infections would be expected to number in the hundreds of millions in the US.

### Limiting mortality is challenging in the context of endemic disease

The suppression of COVID-19 mortality is virtually impossible without eliminating SARS-CoV-2 (Figs 1 and 2). The high expected yearly infection rate for SARS-CoV-2 implies a high disease and mortality burden in the absence of intervention: 159 million infections resulting in 1.1 million deaths could be expected given best-estimates for $R_0$ and the duration of natural immunity (S5 Fig). A vaccine effective against symptomatic disease and mortality but with limited impact on infection and transmission could mitigate such a catastrophic scenario but cannot achieve SARS-CoV-2 suppression or provide communal benefit via "herd immunity."

The impact of such a vaccine on steady-state COVID-19 deaths (Fig 1) is explored. We note that in this parameter range, the duration of immunity is a more significant determinant of

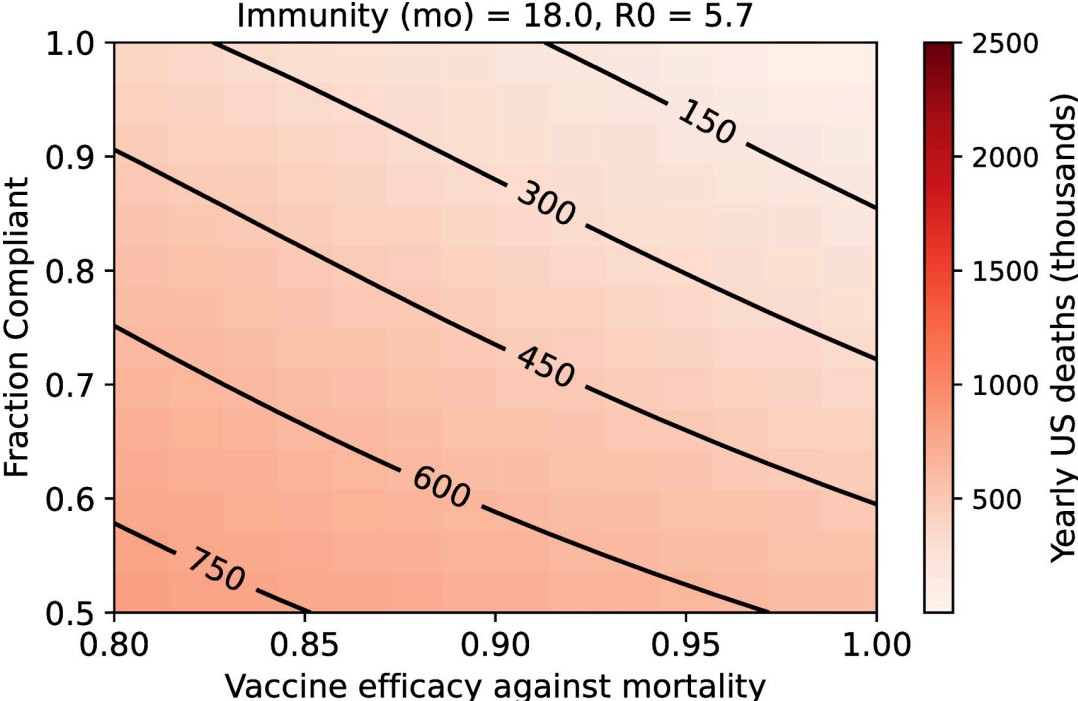

**Fig 1. Steady-state yearly US COVID-19 fatalities after rollout of a vaccine that reduces risk of death ($R_0$ of 5.7 with an 18-month duration of natural immunity).**

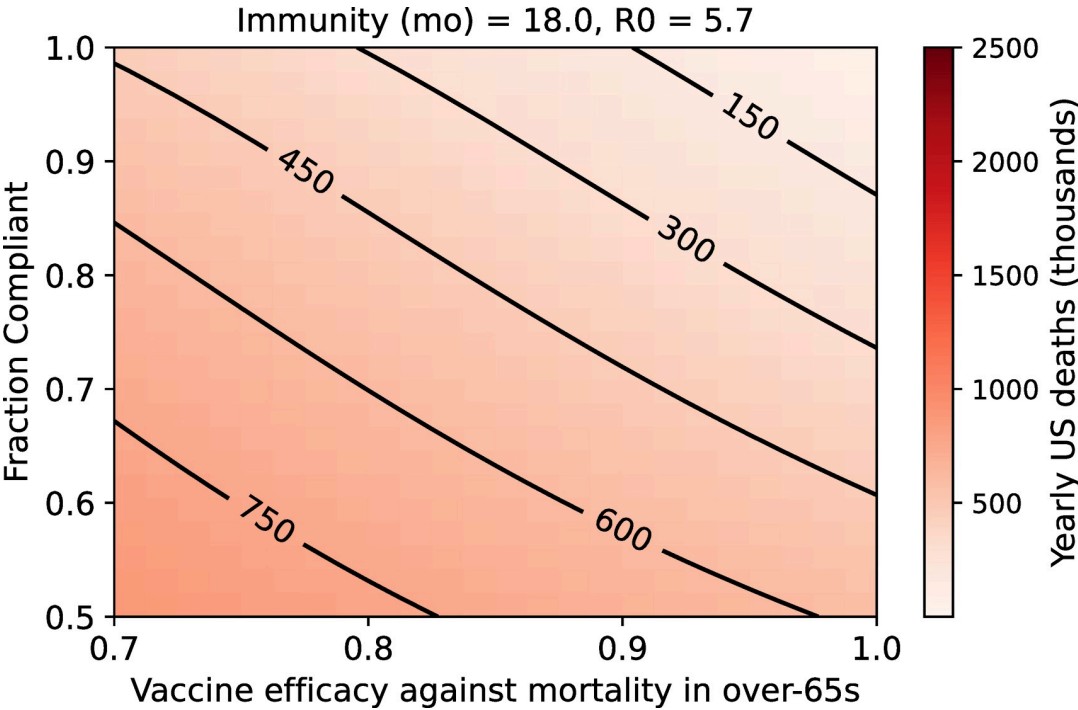

**Fig 2. Range of mortality outcomes in the US population for a vaccine that reduces mortality risk by 95% in the under-65 population, and by varying degrees in the elderly.** (Assumptions are: $R_0 = 5.7$, 18-month immunity).

yearly disease burden than $R_0$ (S1 Fig). As expected, unless a vaccine that is perfectly effective against mortality is administered to all members of the population, COVID-19 deaths will persist indefinitely. Reduced rates of death can be achieved with an effective vaccine, but under pre-pandemic social and economic conditions, extremely high levels of vaccine effectiveness and population compliance are required to achieve "favorable" outcomes (for example, outcomes in which COVID-19 fatalities can be expected to be comparable to yearly influenza deaths.) If a vaccine with 97.5% efficacy against mortality is administered to 95% of the US population—a highly optimistic scenario—approximately 100,000 yearly COVID-19 deaths can be expected (Fig 1). Relaxations in compliance or vaccine efficacy from this highly optimistic case result in large mortality burdens. Deaths remain high as compliance approaches 100% in most scenarios; this means that vaccinated individuals remain at risk of negative health outcomes unless vaccine efficacy is perfect.

### Reduced efficacy in the 65-and-older population presents a serious mortality risk

We conducted a sensitivity analysis on vaccine efficacy in the over-65 population in order to better understand the potential impact of age-dependent efficacy. For example, a vaccine that is 95% effective in all age groups and administered to 90% of the population can be expected to control US COVID-19 fatalities to the level of approximately 200,000 yearly (Fig 1). If this vaccine were 95% effective in Americans younger than 65 but only 86.4% effective in older vaccinees—the current estimate for the Moderna vaccine's efficacy against symptomatic disease in this age group [32]—the model-predicted yearly death exceeds 300,000 (Fig 2). Clearly, vaccine efficacy in the over-65 age group is a highly important determinant of yearly COVID-19 mortality, and uncertainty in vaccine efficacy in this vulnerable population must be addressed.

## Suppression of SARS-CoV-2 is possible with a vaccine that is highly effective at preventing infection and widely administered

Suppression of SARS-CoV-2 is attainable with a vaccine that is highly effective against SARS-CoV-2 infection and transmission and with a high degree of vaccine compliance (Fig 3). Suppression results from virtually zero disease transmission at steady-state because vaccinal immunity in the population impedes sustained spread of the virus. The value of $R_0$ defines the combinations of compliance and vaccine efficacy that result in successful suppression, while the duration of natural immunity determines the yearly disease burden if suppression is not achieved (S3 Fig). At an $R_0$ of 5.7, at least 60% efficacy against infection is required (Fig 3), while at an $R_0$ of 3.32, 45% efficacy is required (S3 Fig); in both of these scenarios, for this level of efficacy against infection to suffice, perfect compliance with vaccination is required. For a vaccine that prevents infection and transmission, deviations from perfect compliance can be compensated by improvements in vaccine efficacy, but even if the vaccine is perfectly effective, a minimum of 82% compliance is required to achieve COVID-19 suppression under an $R_0$ of 5.7, and 70% compliance is required if the $R_0$ is 3.32. Suppression of SARS-CoV-2 with an transmission-preventive vaccine represents a stable strategy: a variety of compliance conditions and vaccine efficacies are acceptable. If vaccinal immunity in the population exceeds the threshold for suppression, small changes in vaccine efficacy or compliance can be tolerated without impacting disease spread.

## The vaccine's effect on transmission by vaccinated infected individuals impacts potential for suppression

Clinical trials generally describe the protective efficacy of a vaccine–that is, the likelihood of illness in a vaccinated individual relative to an unvaccinated individual–but rarely measure the

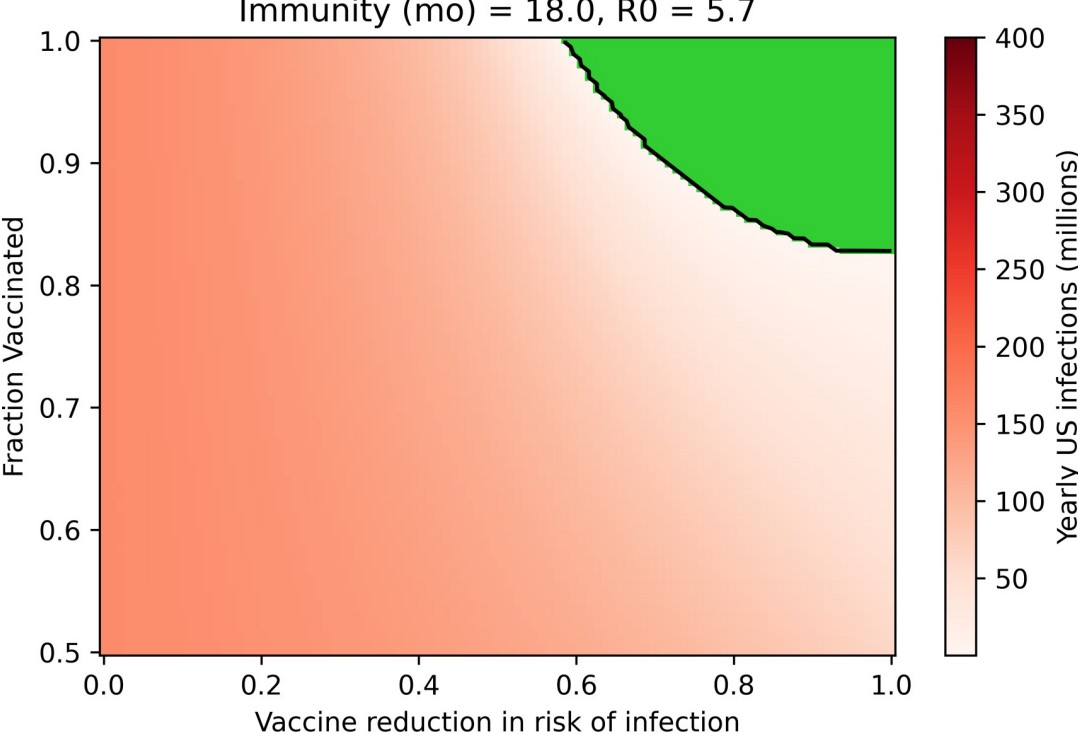

**Fig 3. Suppression of SARS-CoV-2 can be achieved with a highly effective vaccine that is administered to most of the population.** Green region represents cases where suppression is achieved, with virtually zero infections yearly. (Assumptions are: $R_0$ = 5.7, 18-month immunity).

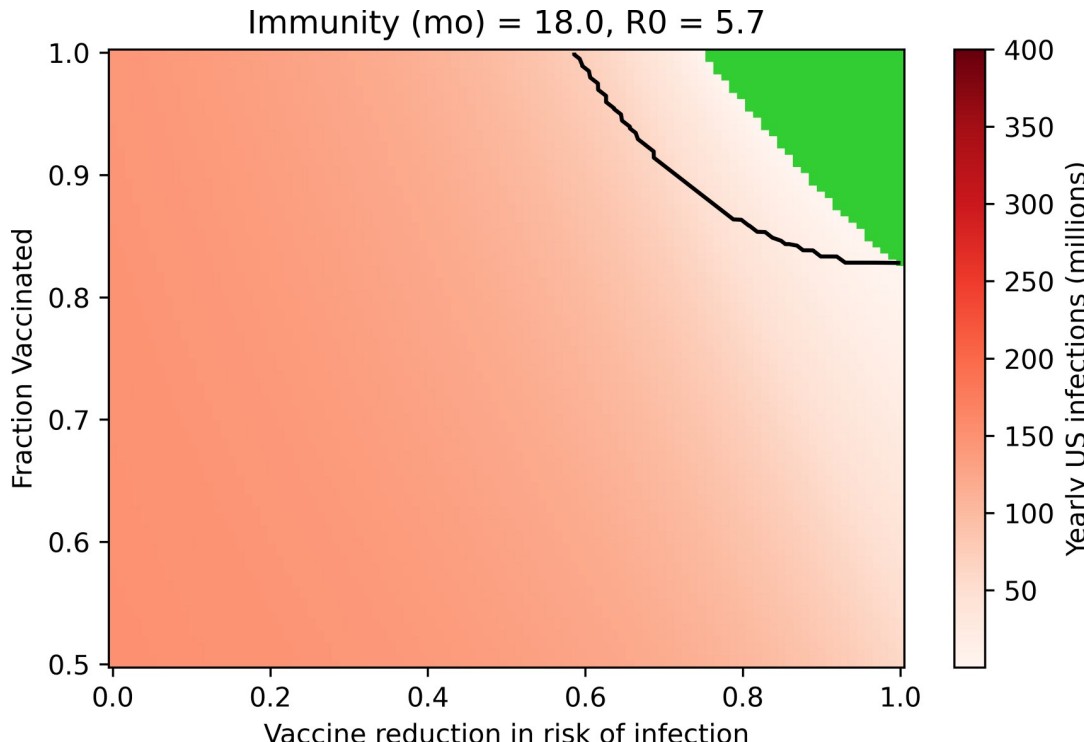

**Fig 4. A vaccine that prevents infection but not transmission requires a higher degree of efficacy and compliance to achieve suppression.** Black lines outline the suppression space for a vaccine that prevents infection and transmission to an equal degree, as shown in Fig 3. Green region represents cases where suppression under the vaccine is achieved, with virtually zero infections yearly. (Assumptions are: $R_0$ = 5.7, 18-month immunity).

difference in likelihood of transmission for an infectious vaccinated individual relative to an infectious unvaccinated individual [47]. In Fig 3, we optimistically assumed that the vaccine's efficacy against transmission in vaccinated infected individuals is equal to its efficacy against infection. However, it is possible that any SARS-CoV-2 vaccine exerts a more limited or non-existent reduction in the likelihood of onward transmission. Fig 4 mirrors Fig 3 while assuming the vaccine has no impact on the likelihood of transmission in infected vaccinated individuals. This change significantly shrinks the available solution space, meaning that a higher degree of compliance and especially vaccine efficacy against infection is required to achieve suppression of SARS-CoV-2. In this case, at least 82% efficacy against infection is necessary if all members of the population are vaccinated and the $R_0$ is 5.7.

S6 Fig further demonstrates the importance of the vaccine's impact on transmission as an aspect of disease suppression strategy: the extent to which a vaccine reduces transmission in vaccinated infected individuals can determine whether the vaccination campaign is successful in eliminating disease. In fact, from a public health perspective, the vaccine's efficacy against onward transmission is as impactful as its efficacy against infection.

## Complementary interventions improve potential for SARS-CoV-2 suppression

Interventions that can be stacked on top of a vaccine, such as masking and antiviral or passive immunity-based prophylaxis, can be deployed to reduce the thresholds for vaccine efficacy and compliance required for SARS-CoV-2 suppression (Fig 5). These measures provide a

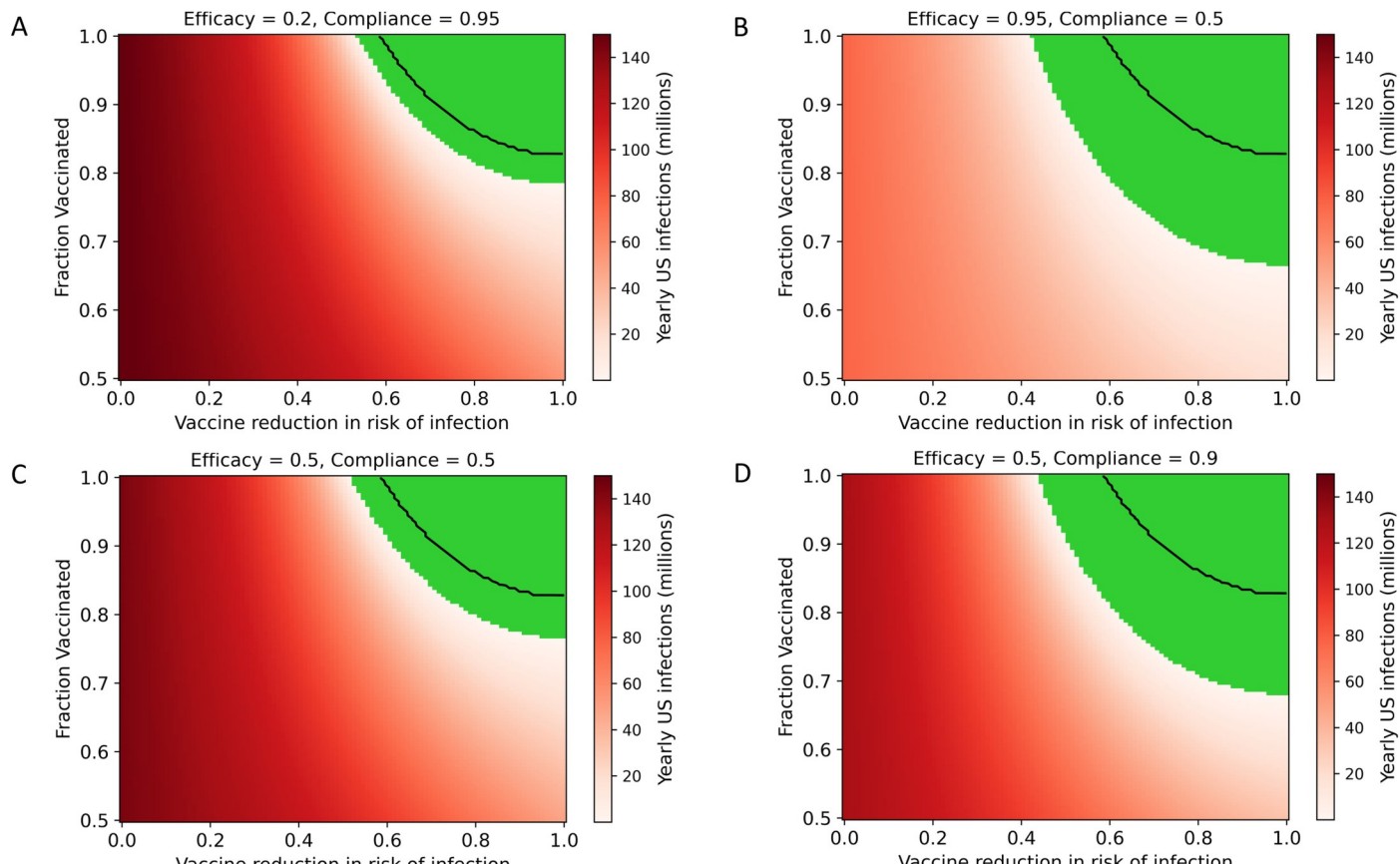

**Fig 5. Interventions complementary to the vaccine improve suppression potential of a vaccine (Assumptions are: $R_0$ = 5.7, 18-month immunity).** Black lines outline the suppression space for the vaccine alone. In the figure panels, four example complementary interventions are explored: A) a passive intervention, such as improving ventilation in schools and workplaces, which impacts 95% of the population but has a small 20% impact on risk of infection; B) a highly effective (95%) intranasal prophylactic that 50% of the population uses; C) a mutually compatible, competing vaccine achieving 50% reduction in risk of infection and 50% compliance in the population; and D) universal masking, which reduces the risk of infection by 50% and reaches 90% compliance.

valuable hedge against risks including imperfect compliance with a vaccine, heterogeneity in regional transmission dynamics, limited vaccine efficacy, changes in vaccine effectiveness due to viral immune escape, waning vaccinal immunity, and unknown vaccinal impact on transmission. Complementary interventions improve the feasibility of suppression across a wide range of possible efficacies and compliance levels. A highly effective (95% [48]) intranasal prophylactic used by 50% of the population, for example, substantially reduces both the compliance and the vaccinal efficacy required to achieve suppression. Universal masking, which can be expected to reduce risk of infection by at least 50% [49], remains a highly effective measure but represents a significant divergence from pre-pandemic activity (the "new normal" as opposed to the "old normal").

## Discussion

In this work, we sought to understand the role that the first wave of SARS-CoV-2 vaccines would play in bringing about a return to normalcy. By definition, a return to normalcy implies a return to pre-pandemic levels of contact, and with this in mind have used an $R_0$ that corresponds to published estimates from the early stages of the pandemic [38]. Our goal in this work was not to predict the future broadly, but rather to focus narrowly on potential

limitations of the first wave of vaccines and to ask "what if" questions around the epidemiological consequences of these potential limitations.

A significant potential limitation of the first wave of SARS-CoV-2 vaccines is the failure to sufficiently prevent infection and transmission in the vaccinated population. While this has not been demonstrated conclusively, it remains a plausible risk that has been observed for other vaccines in the past and that is not being addressed directly in the vast majority of ongoing clinical vaccine studies [6, 32]. In this study, we have used mathematical modeling to understand conditions in the United States at steady state, if a vaccine with limited ability to prevent infection and transmission was used as the primary or sole public-health intervention upon a return to pre-pandemic contact rates. Our results paint a grim picture: at steady state, with a for a vaccine that is 95% effective at reducing mortality and that is taken by 80% of the population, this scenario would lead to 300,000 deaths per year (Fig 1), making COVID-19 the third leading cause of death in the United States [43].

Another significant potential limitation of the first wave of SARS-CoV-2 vaccines is reduced protection for the elderly. This remains a plausible risk as of this writing and would be consistent with the behavior of vaccines for a number of other diseases [29, 31]. We used our modeling framework to explore the impact of a vaccine that was less effective in the elderly. Again, the picture is grim. For example, for a vaccine that is 85% effective in the over-65 population, at steady state, we could expect approximately 450,000 deaths a year (using the assumptions of 95% vaccine efficacy in the under-65 population, 80% compliance, 18 months natural immunity, permanent vaccine immunity and an $R_0$ of 5.7; Fig 2). The vast majority of these deaths would occur in the over-65 population. Beyond the scope of this paper, there are clear implications for the age structure of the US population and life expectancy in general.

The work described here has a number of limitations. First, we assumed that the populations are well-mixed, and did not include stochastic effects. As the focus in this work is examining steady-state mortality and morbidity burdens, this choice is justified as stochastic effects are unlikely to dominate steady-state behavior in a situation with widespread endemic transmission of SARS-CoV-2. Second, we assumed that vaccinal immunity is permanent and time-invariant. The available data for the front-runner vaccines is not consistent with this assumption [5], and waning vaccinal immunity will compound the limitations of real-world vaccine effectiveness described here, further eroding the public-health impact of vaccines that do not prevent transmission. A further limitation of our work is that we focused on the steady-state conditions for the pandemic. As a result, our work does not (seek to) predict the near-term kinetics of COVID-19 within the United States over the next year. We also focused the work narrowly on the United States; while the modeling is generalizable, our mortality and morbidity estimates were focused on US statistics to make the argument easier to understand. We also assumed that stacking interventions (Fig 5) were adopted independently by individuals. A full exploration of the impact of individual choices on the trajectory of the pandemic is beyond the scope of this paper, but has been explored thoroughly by us and others [50, 51]. Lastly, the availability of vaccinal efficacy data limits our analysis. To address this uncertainty, we have performed sweeps to demonstrate the sensitivity to efficacy parameters. In particular, there is insufficient data in the current publications to determine the impact of the vaccines on mortality (in both the Pfizer and Moderna studies, only one COVID-19-related death was reported [8, 32]). The risks of evolved resistance to vaccines, of selection for increasingly transmissible strains of SARS-CoV-2, of waning vaccinal immunity, and of logistical challenges in the induction and maintenance of population-level vaccinal immunity are beyond the scope of this work but layer on top of the risks evaluated in this work.

There are scenarios examined in this paper in which SARS-CoV-2 will become endemic within the population even with effective vaccines as the primary disease-containment

strategy. This outcome has been discussed in the popular press, and is often referred to as "learning to live with the disease" [52, 53]. Such an outcome is in contrast to the "go for zero" [54] approach focused on disease suppression, being pursued by countries such as Australia, New Zealand, China, Taiwan and Korea. Our work shows what "learning to live with the disease" would look like. For the young and healthy, once vaccinated, there may be little incentive to comply with further public health measures aimed at limiting the spread of SARS-CoV-2. If mask-wearing was difficult to incentivize in the population before the deployment of a vaccine, it stands to reason that vaccination will make it more difficult, not easier [51]. For those with limited access to the vaccine or with limited benefit from it, the pandemic will never end.

At a national level, "learning to live with the virus" is a fragile solution that would make the country critically dependent on the vaccine supply chain and vulnerable from a national security perspective to actions by both state and non-state actors. The extensive logistical planning and infrastructure required for maintenance of population vaccinal protection from COVID-19 morbidity and mortality could be readily disrupted by natural disasters, domestic unrest, foreign interference, or future pandemics, resulting in the emergence of symptomatic and deadly COVID-19 in the wake of unrelated crises.

Beyond potential human adversaries, the virus itself is a formidable opponent. In other work [10], we have demonstrated the potential risk of evolutionary immune evasion that comes with strategies that focus on the viral spike protein. Natural selection can be expected to favor increased transmissibility [55], and a transition to increased transmissibility has been shown to occur even after pandemic onset for H1N1 [56, 57] and for SARS-CoV-2 during the current global crisis [58, 59]. Farming SARS-CoV-2 virus at scale with a vaccine that fails to mitigate transmission is an invitation to disaster on this front, as viral evolution over time will limit the space within which a vaccine-mediated strategy can be effective. More transmissible and antibody drug-evading variants of SARS-CoV-2 have already emerged, even before the widespread deployment of antibodies and vaccines [60, 61].

To the extent that vaccines capable of achieving suppression of SARS-CoV-2 exist, failing to achieve suppression is a failure of public policy. If the choice is made to pursue SARS-CoV-2 suppression, vaccines can play a very positive role. Vaccines represent one more layer in the "Swiss cheese strategy" [62–64], as long as it is made clear to vaccinated individuals that they (counterintuitively) may face increased risk after being vaccinated and returning to pre-pandemic lifestyles. To use an analogy, wearing a seatbelt reduces risk of death when driving, but should not be taken as license to down three beers before taking the wheel. Public health messaging should emphasize that the benefit from the vaccine can be offset by actions taken by individuals to increase their contact rate, so compliance with public health measures to reduce transmission will remain in the individual's best interest. Crucially, public health strategy should diversify its approach, broadening focus to passive interventions such as ventilation and complementary biomedical interventions such as intranasal prophylactics. These interventions can be stacked on top of a successful vaccine program to increase the likelihood of eliminating SARS-CoV-2 and to address risks of evolutionary escape, reduced protection in some subpopulations, vaccine noncompliance, and limited efficacy against transmission. Further study of the current vaccines' performance characteristics (impact on infection/ transmission, efficacy in older vaccinees, durability of protection, impact on mortality and morbidity) will be required to better understand their public health impact.

Our work also shows that there is a path forward for vaccines to contribute to SARS-CoV-2 suppression. Vaccines that stop infection and/or transmission can be leveraged to eliminate the disease with even moderately high levels of compliance, a point that has been made by others as well [65]. We submit that the rate of infection and viral load (a predictor of transmissibility [66]) in vaccinated individuals are critical public-health metrics for successful suppression

and may be more meaningful than the number of symptomatic cases. Crucially, stacking interventions (such as indoor air filtration and prophylactics) can help broaden the range of vaccine efficacy and compliance that leads to successful suppression. Complementary vaccines (double vaccinations) may also represent a feasible path forward.

The race to develop vaccines for COVID-19 has yielded an unprecedented technological achievement for mankind, one that would have been inconceivable a decade ago. Martial analogies abound, with the "Manhattan project"-like efficiency of Operation Warp Speed [67], yielding the "weapon that will end the war" [68]. Our work suggests a different military analogy: the bombard cannons of the Hundred Years War promised more firepower than they delivered, and often blew up in the faces of their operators [69–71]. Even so, commanders with a precise understanding of the limitations of these early cannons were able to incorporate them into combined-arms tactics that yielded victories on the battlefield. Clear-eyed rationality about the limitations of these early weapons also allowed for their systematic optimization, and the bombard cannons of the 14th century were iteratively improved until they met their promise and matured into game-changing weapons.

## Supporting information

**S1 Fig. Steady-state yearly US COVID-19 fatalities after rollout of a vaccine that reduces risk of death.** This figure is parallel to Fig 1 in the main text but explores four sets of parameters for the duration of natural immunity and $R_0$. Panels represent four possible scenarios: A) $R_0$ of 3.32 with an 18-month duration of natural immunity, B) $R_0$ of 5.7 with an 18-month duration of immunity, C) $R_0$ of 3.32 with a 6-month duration of immunity, D) $R_0$ of 5.7 with a 6-month duration of immunity.
(TIF)

**S2 Fig. Steady-state US yearly deaths after deployment of a vaccine that reduces mortality risk by 95% in the under-65 population and by varying degrees in the elderly.** This figure is parallel to Fig 2 in the main text but explores four sets of parameters for the duration of natural immunity and $R_0$.
(TIF)

**S3 Fig. Steady-state US yearly SARS-CoV-2 infections after deployment of a vaccine that reduces risk of infection and transmission.** This figure is parallel to Fig 3 in the main text but explores four sets of parameters for the duration of natural immunity and $R_0$. Green region represents regime in which SARS-CoV-2 is eliminated in the population and yearly infections approach zero.
(TIF)

**S4 Fig. Steady-state US yearly SARS-CoV-2 infections after deployment of a vaccine that reduces risk of infection but not transmission.** This figure is parallel to Fig 4 in the main text but explores four sets of parameters for the duration of natural immunity and $R_0$. Black lines outline the suppression space for a vaccine that prevents infection and transmission to an equal degree, as shown in S3 Fig. Green region represents cases where suppression under this vaccine is achieved, with virtually zero yearly infections.
(TIF)

**S5 Fig. Natural herd immunity is no path to normalcy.** US yearly infections (A) and COVID-19 deaths (B) are predicted at steady-state for a variety of $R_0$ and duration of natural immunity estimates. Extensive disease and mortality burdens are expected under all endemic

scenarios.
(TIF)

**S6 Fig. Impact of vaccine reduction in risk of infection and transmission on yearly US infections given 90% vaccine uptake in the population.** Vaccine efficacy against transmission is equally impactful compared to vaccine efficacy against infection and determines success or failure to achieve eradication in many cases.
(TIF)

**S7 Fig. Relationship between vaccine efficacy against infection and transmission and US mortality.** In this figure, 90% of Americans are assumed to be vaccinated and the vaccine is assumed to have an age-independent 95% efficacy against mortality.
(TIF)

**S8 Fig. Relationship between vaccine efficacy against infection and transmission and US mortality.** In this figure, 70% of Americans are assumed to be vaccinated and the vaccine is assumed to have an age-independent 95% efficacy against mortality. At this $R_0$, disease suppression is impossible with only 70% vaccine compliance.
(TIFF)

**S9 Fig. An eradication strategy involving a vaccine that prevents infection but not transmission is more likely to succeed if complementary interventions are in place.** Black lines outline the eradication space for the vaccine alone. This figure is parallel to Fig 4; the vaccine is assumed to have no effect on transmission, the $R_0$ is assumed to be 5.7, and the duration of natural immunity is 18 months. Black lines outline the eradication space for the vaccine alone. In the figure panels, four example complementary interventions are explored: A) a compatible, competing vaccine achieving 50% reduction in risk of infection and 50% compliance in the population; B) universal masking, which reduces the risk of infection by 50% and reaches 90% compliance; C) a passive intervention, such as improving indoor ventilation, which impacts 95% of the population but has a small 20% impact on risk of infection; and D) a highly effective (95%) intranasal prophylactic that 50% of the population uses.
(TIF)

**S10 Fig. Influenza is eradicated more readily than SARS-CoV-2.** Green region represents successful vaccine-based eradication based on an SEIR model for influenza [72].
(TIF)

**S1 File.**
(DOCX)

## Author Contributions

**Conceptualization:** Madison Stoddard, Ryan P. Nolan, Douglas E. White, Natasha S. Hochberg, Arijit Chakravarty.

**Data curation:** Sharanya Sarkar.

**Formal analysis:** Madison Stoddard.

**Supervision:** Arijit Chakravarty.

**Writing – original draft:** Madison Stoddard, Arijit Chakravarty.

**Writing – review & editing:** Madison Stoddard, Sharanya Sarkar, Lin Yuan, Ryan P. Nolan, Douglas E. White, Laura F. White, Natasha S. Hochberg, Arijit Chakravarty.

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
