## [Decision Letter · Decision Letter 0]

22 Apr 2021

PONE-D-21-04641

Beyond the new normal: assessing the feasibility of vaccine-based elimination of SARS-CoV-2

PLOS ONE

Dear Dr. Stoddard,

Thank you for submitting your manuscript to PLOS ONE. After careful consideration, we feel that it has merit but does not fully meet PLOS ONE’s publication criteria as it currently stands. Therefore, we invite you to submit a revised version of the manuscript that addresses the points raised during the review process.

I have received the comments of the reviewers on your manuscript. The specific comments of the reviewers are included below. Please provide point by point response in your revised manuscript.

We look forward to receiving your revised manuscript.

Kind regards,

Muhammad Adrish, MD, MBA, FCCP, FCCM

Academic Editor

PLOS ONE

Journal Requirements:

"I have read the journal's policy and the authors of this manuscript have the following competing interests: MS, LY, and AC are employees and shareholders of Fractal Therapeutics; RN and DW are also shareholders in Fractal Therapeutics."

We note that one or more of the authors are employed by a commercial company: Fractal Therapeutics, Halozyme Therapeutics, Independent Researcher.

2.1. Please provide an amended Funding Statement declaring this commercial affiliation, as well as a statement regarding the Role of Funders in your study. If the funding organization did not play a role in the study design, data collection and analysis, decision to publish, or preparation of the manuscript and only provided financial support in the form of authors' salaries and/or research materials, please review your statements relating to the author contributions, and ensure you have specifically and accurately indicated the role(s) that these authors had in your study. You can update author roles in the Author Contributions section of the online submission form.

2.2. Please also provide an updated Competing Interests Statement declaring this commercial affiliation along with any other relevant declarations relating to employment, consultancy, patents, products in development, or marketed products, etc.  

Reviewers' comments:

Reviewer's Responses to Questions

**Comments to the Author**

1. Is the manuscript technically sound, and do the data support the conclusions?

Reviewer #1: Partly

Reviewer #2: Yes

Reviewer #3: Yes

2. Has the statistical analysis been performed appropriately and rigorously? 

Reviewer #1: Yes

Reviewer #2: I Don't Know

Reviewer #3: N/A

3. Have the authors made all data underlying the findings in their manuscript fully available?

Reviewer #1: Yes

Reviewer #2: Yes

Reviewer #3: Yes

4. Is the manuscript presented in an intelligible fashion and written in standard English?

Reviewer #1: Yes

Reviewer #2: Yes

Reviewer #3: Yes

5. Review Comments to the Author

Reviewer #1: Several comments,

I. Based on the experience of flu vaccines, it is impossible to eliminate zoonotic viral infection in humans with vaccination alone. Therefore, other strategies will be necessary to control the transmission of viral infection but not to eliminate the virus.

II. The R0 represents the number of secondary cases generated by the presence of one infected individual in an otherwise fully susceptible, well-mixed population. In a real world, however, this value changes dynamically with respect to vaccination and/or natural infection status of the population. Therefore, using a fixed value to predict the efficacy or duration of immunity may be inappropriate.

III. As the authors described, vaccination per se can reduce the proportion of symptomatic individuals and even transmission. All available vaccines are administrated intramuscularly and mainly induce systemic immunity but less mucosal immunity. Therefore, it is not surprising that vaccinated persons still get infection, either symptomatic or asymptomatic. This means that other strategies of infection control will be required to stop infection from happening.

IV. Based on the experience of flu vaccines, again, the efficacy of vaccination and duration of immunity will be compromised in the elderly and other immunocompromised patients. Thus, using a fixed value to represent whole population is inappropriate.

V. The references should be updated.

Reviewer #2: The work by Stoddard et al. aims to investigate the role of anti-SARS-CoV-2 vaccines on the possibility of infection eradication hypothesizing different real-life scenarios. Particularly, the authors focus on the possibility of a low protection offered by vaccines towards asymptomatic colonization and the possibility of a reduced effectiveness of vaccines in the eldest people, showing that disease will continue to spread and mortality will be non-negligible. Finally, the authors argue that SARS-CoV-2 vaccines alone won’t allow for a return to the pre-pandemic “normal” and that their use should be done together with complementary public health interventions.

The paper is well written and clear. The outcomes are not generalizable because authors take into account only the American epidemiology, however they specify this limitation in the Discussion section. Also, they cite other important limitations that will require further clarifications in following studies.

Overall, I think that this study is important from a scientific perspective as well as to mitigate enthusiasms to vaccines spread, highlighting the importance of people’ adherence to both vaccination strategies and to other important interventions.

Reviewer #3: Trough epidemiological modeling, the paper by Stottard et al. examined the potential limitation of the first wave of vaccines and the epidemiological consequences of these limitations. The authors critically explored the impact of vaccines on COVID-19 mortality burden. They observed that the complete elimination of SARS-CoV-2 would be possible only with a vaccine that prevents infection and transmission. Although the scenarios described were grim, the main strength of the paper is a good description of the principal limitations of the first wave of vaccines, supported by the mathematical modeling proposed. The authors crucially emphasize the importance of adopting complementary protective measures to reduce transmission between individuals and for return to pre-pandemic lifestyle. Therefore, point out that the current vaccination add one more protective layer in a “Swiss cheese model” of COVID-19 pandemic defense, but it is necessary to increase the vaccine protection from infection and transmission for the SARS-CoV-2 eradication. Methods are adequate; conclusions are supported by the data. The scenario described in this manuscript suggest that if the front-runner vaccines do not prevent infection, SARS-CoV-2 will be endemic in the population and deaths will remain at high levels. This scenario is in contrast with the go for zero policy adopted by country as Australia, China, etc. It would be suitable to Clarify this point

6. PLOS authors have the option to publish the peer review history of their article (what does this mean?). If published, this will include your full peer review and any attached files.

Reviewer #1: No

Reviewer #2: No

Reviewer #3: No

---

## [Author Response · Author response to Decision Letter 0]

30 Apr 2021

Reviewer #1: 

I. Based on the experience of flu vaccines, it is impossible to eliminate zoonotic viral infection in humans with vaccination alone. Therefore, other strategies will be necessary to control the transmission of viral infection but not to eliminate the virus.

The CDC defines elimination as “Reduction to zero of the incidence of a specified disease in a defined geographical area as a result of deliberate efforts; continued intervention measures are required” [1], which is consistent with the reviewer’s statement about “controlling the transmission of viral infection”. However, we agree with the reviewer’s preference for removing the word “eliminate”, as it can be somewhat confusing. We have thus used the term “suppress…transmission” throughout the manuscript to make what we are proposing more clear, in accordance with the reviewer’s request. We further specified that suppression of disease among the human population is the outcome of interest (line 176). 

II. The R0 represents the number of secondary cases generated by the presence of one infected individual in an otherwise fully susceptible, well-mixed population. In a real world, however, this value changes dynamically with respect to vaccination and/or natural infection status of the population. Therefore, using a fixed value to predict the efficacy or duration of immunity may be inappropriate.

We agree with the reviewer on this point. Assessing the impact of changes in R0 on the effectiveness of vaccine-based suppression of SARS-CoV-2 strategy is explored by us in Supplemental Figures S1-S4. The limited durability of natural immunity is varied by us to understand the impact of this parameter on the public health outcomes discussed here (see Figures S1-S4). For the purposes of this manuscript, we chose an (optimistic) assumption where vaccinal immunity did not wane, assumed to be a set of circumstances where individuals are vaccinated often enough that there is no loss of vaccinal immunity. The impact of waning vaccinal immunity on public health outcomes is a very interesting topic, but one that lies outside the scope of this paper. 

III. As the authors described, vaccination per se can reduce the proportion of symptomatic individuals and even transmission. All available vaccines are administrated intramuscularly and mainly induce systemic immunity but less mucosal immunity. Therefore, it is not surprising that vaccinated persons still get infection, either symptomatic or asymptomatic. This means that other strategies of infection control will be required to stop infection from happening.

We agree with the reviewer on this point. This is explored in depth in Figure 5 and Figure S9, and forms part of our argument for requiring multiple complementary interventions. The impact of vaccine efficacy against infection and transmission on disease burden (infections and mortality) are further explored in Figures S6 and S7. 

IV. Based on the experience of flu vaccines, again, the efficacy of vaccination and duration of immunity will be compromised in the elderly and other immunocompromised patients. Thus, using a fixed value to represent whole population is inappropriate.

We agree with the reviewer on this point. Reduced vaccinal protection in the elderly is explored by us in in depth in Figures 2 and S2, and forms part of our argument explaining why vaccines may still permit concerning public health outcomes despite high levels of efficacy in clinical trial populations.

V. The references should be updated.

We thank the reviewer for pointing this out. Since we submitted this paper, much new information has been revealed about the vaccines. Accordingly, we have updated the references in the manuscript extensively, and removed information that has been superseded by more a more complete picture (such as our assessment of Pfizer and Moderna’s vaccines, lines 97-105 and starting at line 496 in the revised manuscript with tracked changes). We believe that our original assessment of a 50% reduction in transmission for the vaccines was overly pessimistic, as new data supports up to a 90% reduction in risk of infection for the Pfizer and Moderna vaccines [2]. As this number was estimated in a non-immunocompromised and younger (20-65 year-old) population at the three month point, we consider this to be a best-case scenario, and so have changed our assumption of reduction of transmission to 70%, to more closely reflect what may be observed in a population with immunocompromised and elderly population over the course of a full year. This is also in line with estimates for the efficacy of the J&J vaccine [3] under best-case conditions. This new estimate was used to re-simulate Figures 1 and 2. The enclosed Powerpoint (not for publication) shows the old and new simulation results side by side. It is our assessment that this change in parameter values does not alter the basic conclusions of our work. Nonetheless, updating the references in the paper will make it more relevant and we thank the reviewer for this recommendation.

Reviewer #2: 

The work by Stoddard et al. aims to investigate the role of anti-SARS-CoV-2 vaccines on the possibility of infection eradication hypothesizing different real-life scenarios. Particularly, the authors focus on the possibility of a low protection offered by vaccines towards asymptomatic colonization and the possibility of a reduced effectiveness of vaccines in the eldest people, showing that disease will continue to spread and mortality will be non-negligible. Finally, the authors argue that SARS-CoV-2 vaccines alone won’t allow for a return to the pre-pandemic “normal” and that their use should be done together with complementary public health interventions.

The paper is well written and clear. The outcomes are not generalizable because authors take into account only the American epidemiology, however they specify this limitation in the Discussion section. Also, they cite other important limitations that will require further clarifications in following studies.

Overall, I think that this study is important from a scientific perspective as well as to mitigate enthusiasms to vaccines spread, highlighting the importance of people’ adherence to both vaccination strategies and to other important interventions.

We thank the reviewer for their generous assessment of our work. We did not note any required changes in this feedback.

Reviewer #3: 

T(h)rough epidemiological modeling, the paper by Stottard et al. examined the potential limitation of the first wave of vaccines and the epidemiological consequences of these limitations. The authors critically explored the impact of vaccines on COVID-19 mortality burden. They observed that the complete elimination of SARS-CoV-2 would be possible only with a vaccine that prevents infection and transmission. Although the scenarios described were grim, the main strength of the paper is a good description of the principal limitations of the first wave of vaccines, supported by the mathematical modeling proposed. The authors crucially emphasize the importance of adopting complementary protective measures to reduce transmission between individuals and for return to pre-pandemic lifestyle. Therefore, point out that the current vaccination add one more protective layer in a “Swiss cheese model” of COVID-19 pandemic defense, but it is necessary to increase the vaccine protection from infection and transmission for the SARS-CoV-2 eradication. Methods are adequate; conclusions are supported by the data. The scenario described in this manuscript suggest that if the front-runner vaccines do not prevent infection, SARS-CoV-2 will be endemic in the population and deaths will remain at high levels. This scenario is in contrast with the go for zero policy adopted by country as Australia, China, etc. It would be suitable to Clarify this point

We thank the reviewer for their generous assessment of our work. Based on their recommendation, we have made the recommendation of using vaccines for suppression more explicit in the discussion (lines 538-539 in the revised manuscript with tracked changes).

References

1. The Principles of Disease Elimination and Eradication. [cited 2021 Apr 30]. Available from: https://www.cdc.gov/mmwr/preview/mmwrhtml/su48a7.htm

2. Thompson MG. Interim Estimates of Vaccine Effectiveness of BNT162b2 and mRNA-1273 COVID-19 Vaccines in Preventing SARS-CoV-2 Infection Among Health Care Personnel, First Responders, and Other Essential and Frontline Workers — Eight U.S. Locations, December 2020–March 2021. MMWR Morb Mortal Wkly Rep. 2021;70. Available from: https://www.cdc.gov/mmwr/volumes/70/wr/mm7013e3.htm

3. GRADE: Janssen COVID-19 Vaccine | CDC. 2021 [cited 2021 Apr 27]. Available from: https://www.cdc.gov/vaccines/acip/recs/grade/covid-19-janssen-vaccine.html

---

## [Decision Letter · Decision Letter 1]

2 Jul 2021

Beyond the new normal: assessing the feasibility of vaccine-based suppression of SARS-CoV-2

PONE-D-21-04641R1

Dear Dr. Stoddard,

We’re pleased to inform you that your manuscript has been judged scientifically suitable for publication and will be formally accepted for publication once it meets all outstanding technical requirements.

Kind regards,

Muhammad Adrish, MD, MBA, FCCP, FCCM

Academic Editor

PLOS ONE

Additional Editor Comments (optional):

All comments have been addressed.

Reviewers' comments:

Reviewer's Responses to Questions

**Comments to the Author**

1. If the authors have adequately addressed your comments raised in a previous round of review and you feel that this manuscript is now acceptable for publication, you may indicate that here to bypass the “Comments to the Author” section, enter your conflict of interest statement in the “Confidential to Editor” section, and submit your "Accept" recommendation.

Reviewer #2: All comments have been addressed

Reviewer #4: All comments have been addressed

2. Is the manuscript technically sound, and do the data support the conclusions?

Reviewer #2: Yes

Reviewer #4: Yes

3. Has the statistical analysis been performed appropriately and rigorously? 

Reviewer #2: Yes

Reviewer #4: Yes

4. Have the authors made all data underlying the findings in their manuscript fully available?

Reviewer #2: Yes

Reviewer #4: Yes

5. Is the manuscript presented in an intelligible fashion and written in standard English?

Reviewer #2: Yes

Reviewer #4: Yes

6. Review Comments to the Author

Reviewer #2: (No Response)

Reviewer #4: This paper presents some modelled outcomes of different vaccination scenarios. Some of these elements are playing out in real time and new data are becoming available while this paper is under review. Consequently by the time this is published some of these scenarios might not be something we will actually face. However, as a modeling exercise, there is still something to be learned from the paper. I feel the authors have adequately addressed the comments made by previousl reviewers.

7. PLOS authors have the option to publish the peer review history of their article (what does this mean?). If published, this will include your full peer review and any attached files.

Reviewer #2: No

Reviewer #4: No

---

## [Editor Report · Acceptance letter]

8 Jul 2021

PONE-D-21-04641R1 

Beyond the new normal: assessing the feasibility of vaccine-based suppression of SARS-CoV-2 

Dear Dr. Stoddard:

I'm pleased to inform you that your manuscript has been deemed suitable for publication in PLOS ONE. Congratulations! Your manuscript is now with our production department. 

Kind regards, 

on behalf of

Dr. Muhammad Adrish 

Academic Editor

PLOS ONE